# Comprehensive analysis of effect of high-acid-value waste cooking oil on adhesion between aged asphalt and aggregates

Zhiyu Wang[1,2], Qiang Pei[3¤b], Kunjie Li[4¤c], Zhonghui Wang[1,2], Xiaodong Huo[1,2], Yongwei Wang[1,2], Yanfang Li[1,2¤a], Shaoqi Kong[5¤d]*

**1** Department of Chemical and Materials Engineering, Lyuliang University, Lyuliang, Shanxi, China, **2** Institute of New Carbon-Based Materials and Zero-Carbon and Negative-Carbon Technology, Lyuliang University, Lyuliang, Shanxi, China, **3** National and Local Joint Engineering Laboratory of Advanced Road Materials, Taiyuan, Shanxi, China, **4** National Key Laboratory of Coal and Coalbed Methane Mining, Taiyuan, Shanxi, China, **5** College of Mining Engineering, Taiyuan University of Technology, Taiyuan, Shanxi, China

¤a Current Address: Department of Chemical and Materials Engineering, Lyuliang University, Lyuliang. Lyuliang, China
¤b Current Address: National and local joint engineering laboratory of advanced road materials, Taiyuan, Shanxi Province, China
¤c Current Address: National and local joint engineering laboratory of advanced road materials, Taiyuan, Shanxi Province,China
¤d Current Address: College of Mining Engineering, Taiyuan University of Technology, Taiyuan, Shanxi Province, China
* kongshaoqi@tyut.edu.cn

## Abstract

Waste cooking oil (WCO) has widespread applications in the rejuvenation of aged asphalt. A few investigations have been conducted on the multiscale adhesion properties and rejuvenation mechanism of aged asphalt-aggregate surfaces treated with high-acid-value WCO. This research focuses on the implications of WCO on the interfacial adhesion of aged asphalt from both macroscopic and microscopic structural viewpoints. The asphalt–aggregate interfacial adhesion was macroscopically assessed through binder bond strength (BBS) tests, and the microscopic mechanism of activation was investigated using a molecular dynamics model. The findings show that through promoting microstructure recovery and mitigating the adverse consequences of aging, WCO might enhance the adhesion strength of aged asphalt. Although the effects of aging on asphalt interface adhesive depend on mineral category, WCO could enhance adhesion along with moisture resistance over the aged asphalt-aggregate boundary. Furthermore, binding strength and interface adhesion work have a strong positive association. This research provides an exhaustive knowledge of the adhesive characteristics of the asphalt-aggregate interface at the multiscale level.

**Data availability statement:** The data are held or will be held in a public repository: 10.57760/sciencedb.28482.

**Funding:** The authors acknowledge support from the Luliang City High-Level Scientific and Technological Talent Introduction Key Research and Development Project (2022RC17), Luliang City high-tech field key research and development projects (2024GX06), Luliang City high-tech field key research and development projects (2024GX07), Shanxi Province Science Foundation for Youths Award (202303021222253), and the Innovative Development Plan Project of Shanxi Transportation Research Institute Group Co., Ltd. (22-JKCF-07).

**Competing interests:** The author declares that there are no conflicts of interest.

## 1. Introduction

The anticipated practical use of high-content recycled asphalt pavement (RAP) in road construction and maintenance is increasing due to its economic, social, and environmental advantages [1,2]. The colloid structure and viscoelasticity of asphalt deteriorate over time due to chemical transformation [3]. Rejuvenators activate aged asphalt as well as alleviate a detrimental impact of aging, potentially encouraging mixing of primary and aged asphalt while also enhancing RAP content and performance [4–7]. Waste cooking oil (WCO) is identical to the light components in asphalt and serves as a sustainable rejuvenation agent for aged asphalt [8]. The degree of bonding between asphalt binders and mineral aggregates is critically important for crack resistance [9], moisture resistance, and structural integrity to asphalt-aggregate composites, which ultimately determine the long-term durability of pavements [10,11]. Adhesion is predominantly caused by the micro-chemical and physical interactions between the connected surfaces [12]. The interfacial adhesion mechanism of composite materials should be studied at the molecular level to improve their macroscopic properties [13]. Specifically, the interfacial features of WCO-rejuvenated asphalt should be investigated extensively to advance the use of RAP substances in roadway construction.

Binder bond strength (BBS) tests are widely used to evaluate the adhesion performance of asphalt through simple, quick, accurate measurement [14]. However, these tests only facilitate visual evaluation of interfacial adhesion; they do not provide an adequate understanding of the microscopic mechanisms underlying the performance evolution of asphalt mixtures [15]. Molecular dynamics (MD) provides advantages to examining the interfacial properties of asphalt mixtures and asphalt dispersion on mineral surfaces at the molecular level [16]. Gao [17] et al. investigated the roles of mineral types and moisture on adhesion and debonding work, figuring out that alkaline minerals bond better than acidic minerals and that water impedes mineral adhesion. According to Chu [18] et al., the asymmetry of aggregate textures has an extensive impact on aggregate-asphalt binding strength. Gao [19] et al. observed that oxidation affects interfacial adhesion with regard to mineral acidity and alkalinity. As stated by Sun [20] et al., water influences asphaltene self-aggregation and the distribution of asphalt components. Long [21] et al. evaluated the interfacial bonding mechanism of a nanosilica-modified asphalt mixture. According to Han [12] et al., rejuvenating agents strengthen the adhesion features of aged asphalt through regulating their components and replenishing lightweight components lost with age. Yan [22] et al. investigated the consequences of WCO on asphalt-aggregate adhesion, revealing that WCO enhances resistance to damage by moisture. The previous study concentrated on the effect of WCO on the adhesion properties of aged asphalt and aggregates at the macroscopic or microscopic scales. However, the implementation of multi-scale methodologies for investigating the regenerative effect of long-term use of high acid value WCO on aged asphalt adherence remains limited, and the nature of the regeneration procedure remains unknown.

The primary goal of this study was to identify the restoration effect and rejuvenation mechanisms of high-acid-value WCO on the adhesive characteristics of aged

asphalt at multiscale levels. Macroscopic experiments may validate microscopic research, whereas microscopic study explore the mechanisms that underlying macroscopic activities. Pull-off tests were initially implemented to examine the macroscopical consequence of WCO on asphalt-aggregate adhesion. Asphalt-aggregate adhesion and debonding characteristics were quantitatively analyzed using MD-simulated adhesion and debonding work. The microscopic mechanism of WCO's rejuvenation effect on aged asphalt was investigated using the radial distribution function (RDF), mean square displacement (MSD), and relative concentration function (RCP). All experiments included diverse aggregate types and alterations in WCO content. The findings can be used to assess the effect of WCO on the interface systems of aged asphalt from various viewpoints, thoroughly elucidate its mechanism of action, and provide foundational information for the effective use of RAP materials.

## 2. Materials and methods

### 2.1. Material preparation

The primary material in this investigation was Klamayi 70# asphalt, and the aged asphalt was manufactured in accordance with JTG E20-2011 standards. The essential technical data parameters were displayed in Table 1.

Table 2 displays the characteristics of WCO that were commonly employed by fried chicken restaurants.

Rejuvenated asphalt was produced through mixing varying amounts of WCO (3%, 6%, and 9%) to aged asphalt and whirling the mixture for 20 minutes at 4000 rpm at approximately 135°C. Granite and limestone, which are commonly used stones in domestic road engineering, were utilized to assess the bonding capacity at the interfaces of rejuvenated asphalt and aggregates. Table 3 details the characteristics of the limestone and granite used here.

### 2.2. Experiment methods

Pull-off tests were performed utilizing a DeFelsko PosiTest AT-M adhesion tester at a loading rate of 0.5 MPa/s and a temperature of 25±1°C in compliance with ASTM D4541-17 [23]. Five parallel experiments were carried out under each

**Table 1. Features form asphalt.**

| Index | Test method | Primary asphalt | Aged asphalt |
|---|---|---|---|
| Penetration (25°C, 0.1 mm) | ASTM D5 | 68.2 | 40.5 |
| Softening point (°C) | ASTM D36 | 49.2 | 64.3 |
| Ductility (15°C, cm) | ASTM D113 | 105.7 | 90.3 |

**Table 2. Properties of WCO.**

| Index | Test method | Value |
|---|---|---|
| Density (25°C, g/cm³) | T0603 | 0.91 |
| Acid value (mgKOH/g | T0626 | 12.2 |
| Viscosity (25°C, Pa·s) | T0625 | 0.05 |

**Table 3. Fundamental characteristics of aggregates.**

| Index | Test method | Granite | Limestone |
|---|---|---|---|
| Bulk density (g/cm³) | ASTM C127 | 2.54 | 2.18 |
| Water absorption (%) | ASTM C128 | 3.16 | 2.07 |
| Roughness (μm) | ASTM D7363 | 10.17 | 5.68 |

asphalt condition, and the test values were averaged. Pull-off test data were mainly applied to clarify the effect of WCO on the interface features of asphalt mixes quantitatively and verify the accuracy of MD modeling.

### 2.3. Molecular structure and simulation methods

**2.3.1. Molecular structure of asphalt and high-acid-content WCO.** In this study, four components (12 molecules) provided by Li and Greenfield [24] were chosen to represent the virgin asphalt and the validity of selecting these components has been well proven [20,25]. Fig 1 represents the molecular structure about virgin asphalt.

Ketones attending benzene carbon atoms as well as sulfoxides onto sulfur atoms were developed as templates for aged asphalt units based on an asphalt aging mechanism [26]. Table 4 illustrates the mass percentages and molecular numbers of each component in the primary and aged asphalt systems. The x and y dimensions of the asphalt layer coincided with the mineral matrix model.

Oleic acid, palmitic acid, and linoleic acid were the main components of high-acid-value WCO [27–29]. In this study, oleic acid and palmitic acid were selected as typical WCO fragments with high acid values, and the three WCO proportions in the model were investigated by altering the quantity of molecules.

**2.3.2. Molecular model of mineral aggregates.** $SiO_2$ and $CaCO_3$ were used to reflect the chemical compositions of popular aggregates [30]. The mineral substrate model was manufactured by dividing the mineral unit cell structure along the [1,0,0] plane and repeating it in the x and y planes of the two-dimensional form. A 50 Å vacuum plate was placed to the surface, forming a mineral block with a three-dimensional periodic boundary.

**2.3.3. Modelling for asphalt-aggregate interactions.** A constrained asphalt layer was installed on the top of the minerals ground for generating the model of the asphalt-aggregate interface model. A thin layer of 200 water molecules was incorporated to the asphalt-aggregate border for generating an asphalt-water-aggregate sample. Fig 2 depicts the asphalt-aggregate interaction structures either with or without water molecules.

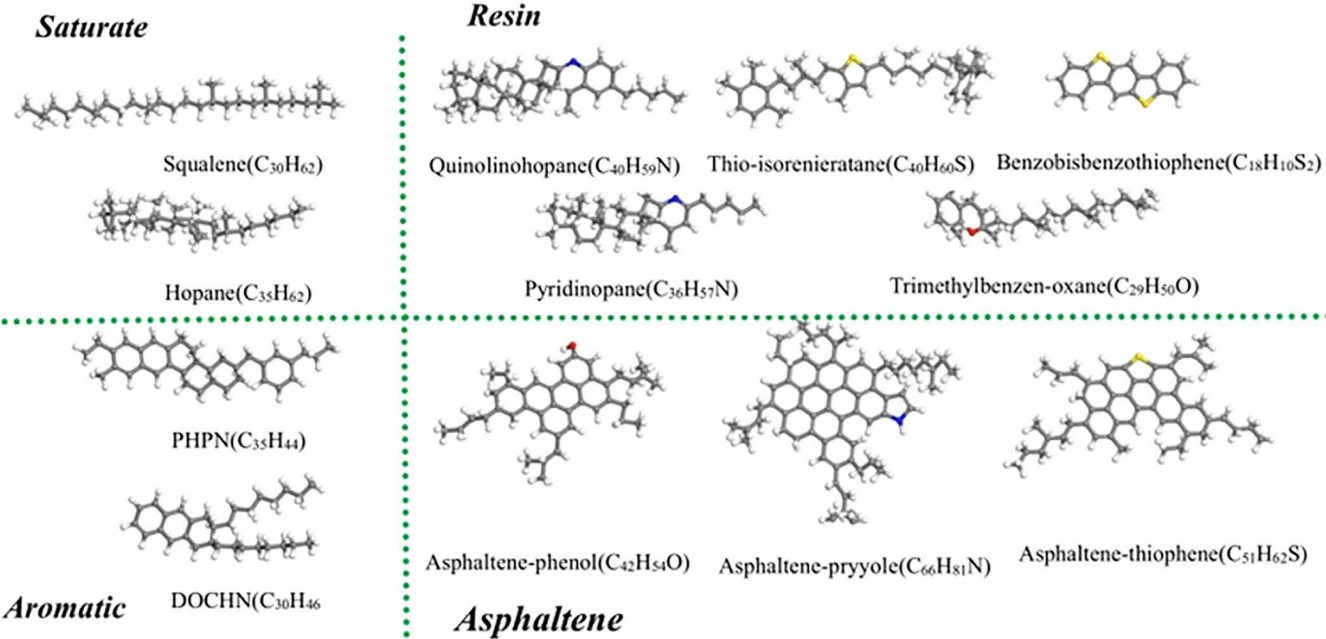

**Fig 1. Twelve-component molecular structure for virgin asphalt saturate, aromatic, resin, and asphaltene (SARA) components (gray for carbon, white for hydrogen, yellow for sulfur, and red for oxygen).**

**Table 4. The combined composition includes virgin and aged asphalt.**

| Component | Model quantity | Primary asphalt | | Aged asphalt | |
|---|---|---|---|---|---|
| | | Molecular structure | Quality fraction (%) | Molecular structure | Quality fraction (%) |
| Saturate (S) | 4 | $C_{30}H_{62}$ | 11.11 | $C_{30}H_{62}$ | 10.32 |
| | 4 | $C_{35}H_{62}$ | | $C_{35}H_{62}$ | |
| Aromatic (A) | 11 | $C_{35}H_{44}$ | 31.90 | $C_{35}H_{36}O_4$ | 32.41 |
| | 13 | $C_{30}H_{46}$ | | $C_{30}H_{42}O_2$ | |
| Resin (R) | 4 | $C_{40}H_{59}N$ | 39.74 | $C_{40}H_{55}O_2N$ | 39.60 |
| | 4 | $C_{40}H_{60}S$ | | $C_{40}H_{56}O_3S$ | |
| | 15 | $C_{18}H_{10}S_2$ | | $C_{18}H_{10}O_2S_2$ | |
| | 4 | $C_{36}H_{57}N$ | | $C_{36}H_{53}O_2N$ | |
| | 5 | $C_{29}H_{50}O$ | | $C_{29}H_{48}O_2$ | |
| Asphaltene (A) | 3 | $C_{42}H_{54}O$ | 17.25 | $C_{42}H_{46}O_5$ | 17.67 |
| | 2 | $C_{66}H_{81}N$ | | $C_{66}H_{67}O_7N$ | |
| | 3 | $C_{51}H_{62}S$ | | $C_{51}H_{54}O_5S$ | |

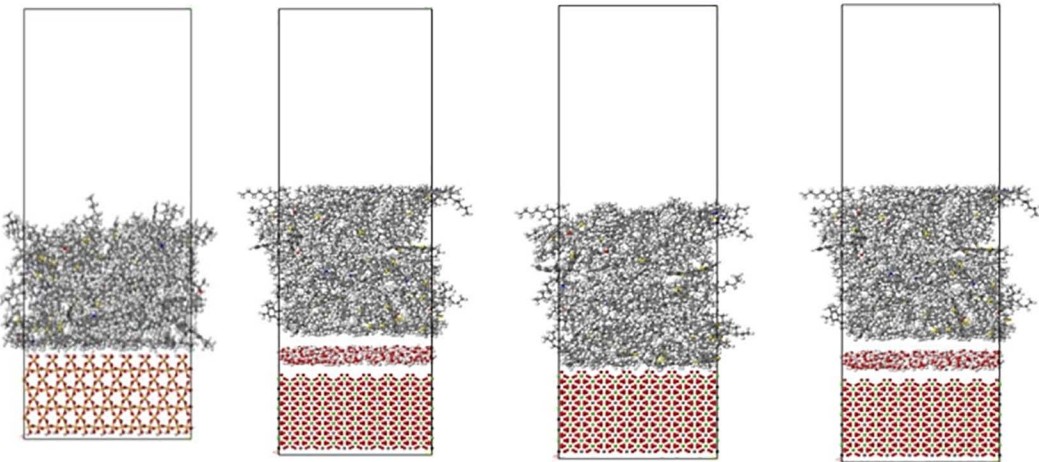

**Fig 2. Asphalt-aggregate mineral interaction models with and without water.**

**2.3.4. Molecular simulation.** The interface models underwent 5000 iterations of geometric transformation, followed by an annealing stage of 5 cycles at 250 ps each, with temperatures that varied between 298–500 K. A 200 ps simulation was then conducted using a canonical ensemble (NVT) to further optimize the model arrangement, followed by a 100 ps simulation for determining the required performance parameters.

Materials Studio 2019 software was adopted in this investigation to generate the model and evaluate its material properties. The COMPASS II force field was selected, that was commonly used in molecular dynamics simulations of asphalt materials [31–33]. The Nosé-Hoover thermostat and Andersen barostat were used during relaxation to maintain a constant temperature and pressure. The van der Waals interaction was derived through atom-based summation with a cutoff length of 15.5 Å. Ewald summation was used to calculate the electrostatic interactions with a threshold radius of 6 Å.

## 3. Results

### 3.1. Macroscopic interface adhesion

The pull-off test has been an excellent method for determining the effectiveness of adhesion in the asphalt-aggregate interaction structure [34]. The consequence of WCO on the interfacial bonding features of aged asphalt was assessed through the measurement of the interface's pull-off strength. Fig 3 displays the binding strengths of different asphalt forms. Since the acidic sections of virgin asphalt displayed better chemical adsorption with the alkaline limestone aggregate, the association was stronger than that with granite. Aging weakened the asphalt-granite adhesion force while strengthening the asphalt-limestone bond. The incorporation of WCO substantially improved the adhesive strengths between the aged asphalt and the aggregates, consequently regenerating the aged asphalt's adhesion characteristics.

### 3.2. MD simulation assessments

#### 3.2.1. Influence of WCO on microscopic structure of asphalt.
The intermolecular structure of asphalt influences its overall physical properties, and has been a focus of research for many years. The RDF represents the probability of recognizing an intended particle within a radius r of a specific particle, corresponding to the degree of aggregation of the pieces [19,35]. The peak value is significant, indicating extensive intermolecular aggregation at that specific location. This function is expressed as Eq. (1).

$$g(r) = \frac{dN}{\rho 4\pi r^2 dr}$$

(1)

where $\rho$ is the mean density of a system, $N$ is the quantity of particles, and $r$ is the separation between particles.

Asphalt binders have been widely demonstrated to have a natural colloidal structure in which asphaltenes were surrounded by resin and dispersed in maltenes composed of saturated and aromatic components. The aggregation behavior of asphaltene, the most viscous and polar component of asphalt, significantly influences asphalt's physical properties [20,36]. As consequence, the implication of WCO regarding the molecular structure properties of aged asphalt was

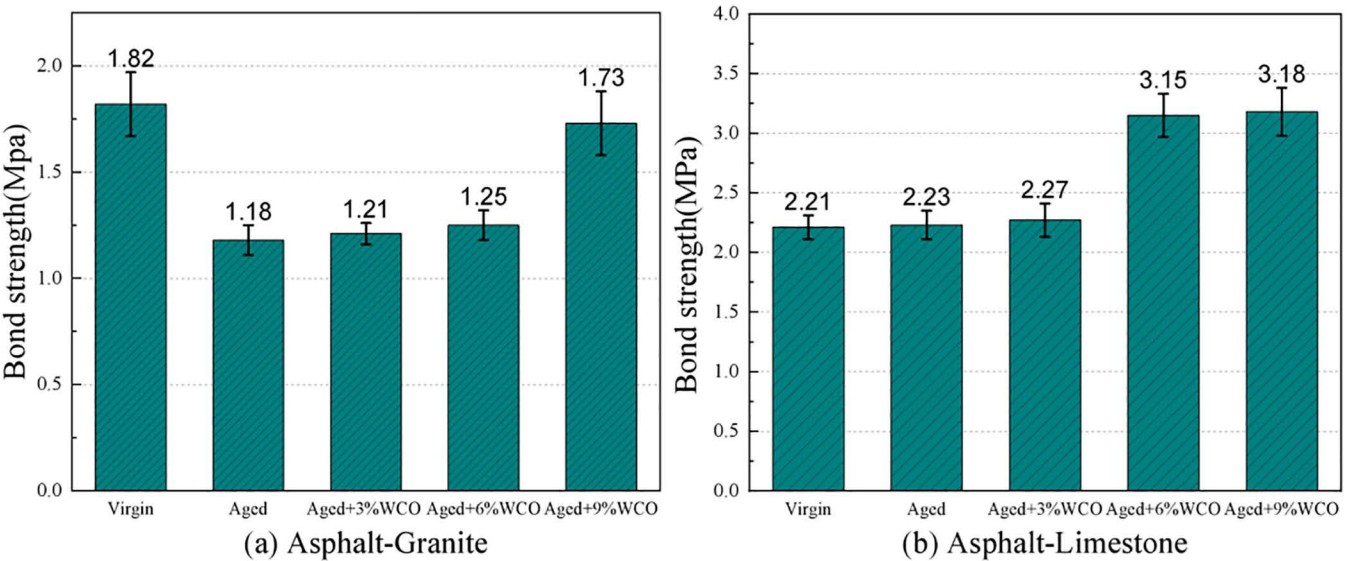

**Fig 3. Strength within asphalt-aggregate interfacial associations.**

explored through the RDF curve of asphaltene-asphaltene pairs, which provides a potential methodology to discover the rejuvenation foundation to aged asphalt.

In this work, the dispersion of asphaltene in various asphalt models was assessed by RDF curves, which the distance gained through the molecular centroid. Fig 4 depicts the RDFs for the asphaltene-asphaltene pair within virgin, aged, and rejuvenated asphalt. The horizontal coordinate reflects the separation among molecules at the truncation point 25 A, whereas the vertical value indicates g(r).

Fig 4 indicates that the aged asphalt displayed significant self-aggregation, as evidenced by its high g(r) and low r, in contrast to the virgin asphalt. The primary cause was that ketone and sulfoxide functional groups were created within asphaltene during the oxidation aging process, which increases polarity and self-polymerization tendency [37]. The intensification of asphaltene accumulation was the main cause of degradation of the properties of the aged asphalt [38,39].

The characteristics of the aggregate interface have substantial effects on the microstructure of asphalt. Asphaltene aggregated more strongly on the $SiO_2$ surface than on the $CaCO_3$ surface. This was due to the strong electrostatic bond of alkaline $CaCO_3$, which drew a portion of the asphaltene from the bulk asphalt, decreasing the asphaltene accumulation on the calcite surface [19]. For the asphalt-$SiO_2$ model, the weak interaction between the acidic minerals and bitumen minimally affected the asphalt aggregation state adjacent to the minerals [40].

While WCO was incorporated, the maximum numbers related to the asphaltene-asphaltene couples decreased and moved to the right side. The decrease in the asphaltene-asphaltene aggregation was positively related to the WCO dosage. It could be acceptable that the WCO molecules reduced the attraction between molecules of the asphaltenes and improved the microscopic structure of the aged asphalt [41] by inhibiting the strong interactions among polarized units. The most effective approach for restoring the original characteristics of aged asphalt represents by means of the depolymerization effect of WCO, which represents a genuine regeneration operation [42,43].

**3.2.2. Influence of WCO on asphalt diffusion.** The role of WCO in the migration characteristics of the asphalt ingredients across the aggregate surfaces was investigated quantitatively using the MSD, which can reflect the dynamic adhesion behavior of asphalt elements and minerals [44]. MSD has been expressed as a square of a particle's displacement during an exact period, as indicated in Eq. (2) [45,46].

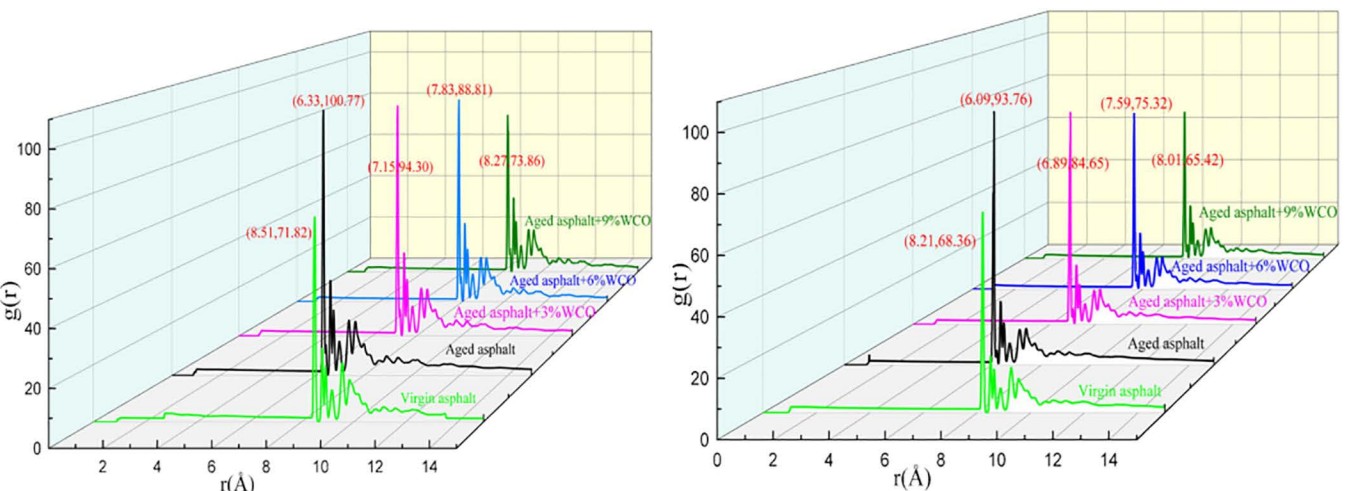

**Fig 4. RDF curves for asphaltene-asphaltene couples under different asphalt models.**

$$MSD(t) = r^2(t) = \frac{1}{N}(\sum_{i=1}^{N}\left|r_i(t) - r_i(0)^2\right|) \tag{2}$$

Where, N represents the number of particles, t denotes the time interval, and ri(t) and ri(0) indicate the radial positions of particle i at time t and the initial time, respectively.

The diffusion coefficient $D$ was determined using a gradient of the MSD trajectory, as described by Eq. (3), where a higher slope denotes greater molecular mobility.

$$D = \frac{1}{6}\lim_{t\to\infty}\frac{d}{dt}\sum_{i=1}^{N}\left\langle\left|r_i(t) - r_i(0)^2\right|\right\rangle \tag{3}$$

Fig 5 depicts the $D$ values for the SARA components under different asphalt conditions. The figure shows a correlation between the molecular mobility of a SARA component and its molecular weight. Saturates with low molecular weights have greater fluidity, whereas asphaltenes with high molecular weights have reduced diffusion ability. Asphalt aging reduces the mobility of SARA components, reflecting the underlying aging mechanism. The primary cause of the decreased saturate mobility was a reduction in the free volume percentage (FFV) [31]. The active function groups within Ar, Re, and As altered or were substituted with age, which caused substantial rises in molecular weights and polarities [47]. In summary, aging strengthened molecular binding and reduced the FFV, which affected the molecular movement and resulted in an overall reduction in $D$ [48].

The incorporation of WCO strengthened the mobility of the aged asphalt. This was attributed towards the decomposition actions of the WCO upon the aged asphalt, just as demonstrated by the RDF analysis results in Section 3.2.1. The degree to which WCO influenced the mobility of the SARA components varied, mainly depending on molecular size [49]. The nature of aggregates significantly affects the mobility of asphalt components. SARA components moved more quickly

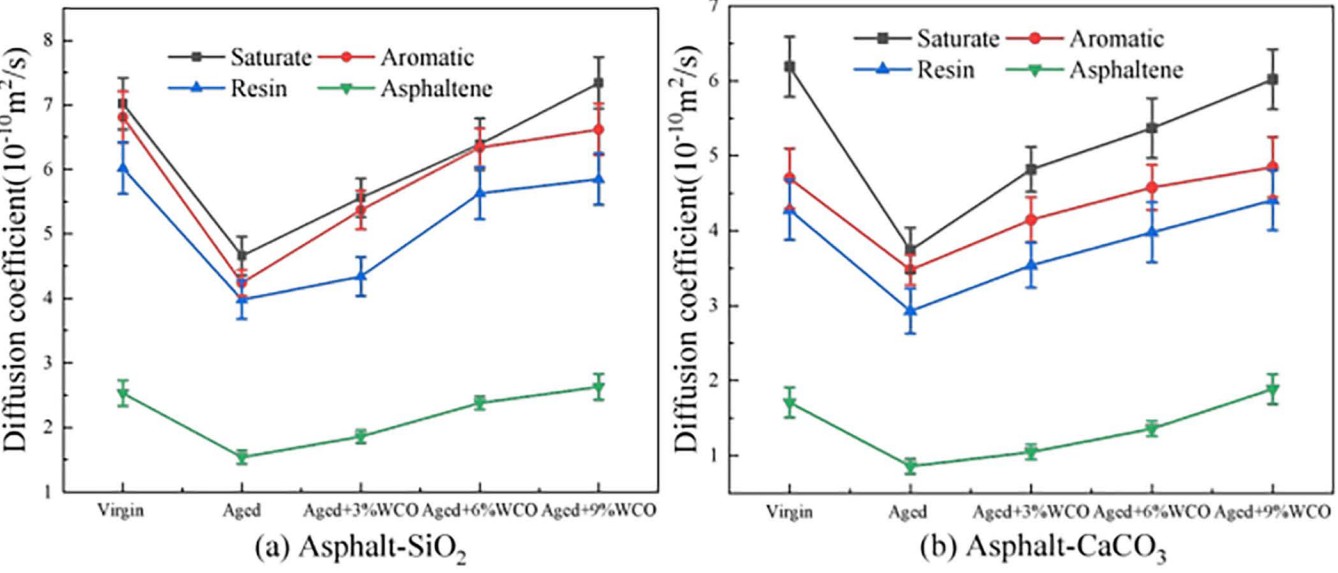

**Fig 5. Diffusion coefficients of SARA components in different asphalt models.**

on SiO$_2$ surfaces than on CaCO$_3$ surfaces, owing to the strong attraction between asphalt and alkaline minerals, which limited the mobility of asphalt molecules [19].

**3.2.3. Influence of WCO on distribution of asphalt components.** An axial content arrangement for SARA parts along the interface was another indicator implemented to accurately evaluate the microstructure of asphalt. Divide the system vertically into 100 compartments, then calculate the relative concentration of each section through dividing the mass density of each compartment by the system's mass intensity [20,50].

Fig 6 illustrates how WCO affected the axial concentration arrangement of the SARA components. The asphalt dispersion on the surface of the aggregates can be recognized as the asphaltenes aggregate to form clusters or micelles, which dispersed in a maltene constituted of saturated, aromatics, and resins. The RCP of Re and As on the CaCO$_3$ surface was greater than that on the SiO$_2$ surface due to the strong connection between the polar elements of the asphalt and the alkaline material. With oxidative aging, the asphalt was distinguished from the aggregates, and the percentage of SARA components over the aggregate interface decreased significantly. This finding aligned with already exhibiting results [18,31].

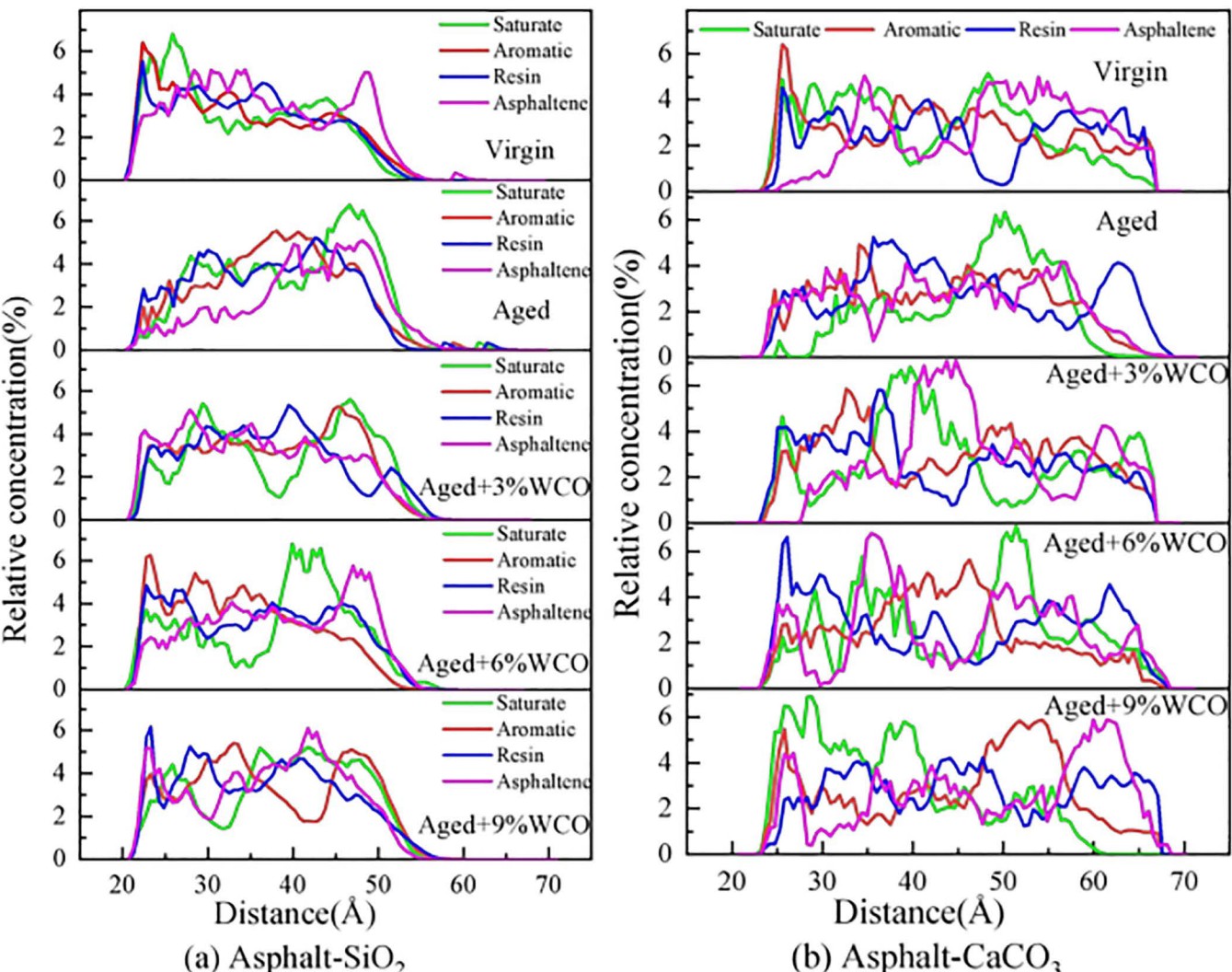

**Fig 6. Impact of WCO on vertically percentage distributions of SARA constituents.**

To a certain degree, the WCO could improve the axial distribution of SARA components within aged asphalt on aggregate surfaces. A higher concentration of lightweight components helped soften the aged asphalt [37]. Variations in component distribution considerably affected the performance features of the asphalt.

**3.2.4. Influence of WCO on asphalt-aggregate adhesion.** The mechanical properties in asphalt mixtures were mainly determined by the interfacial interaction of the two fundamental components, asphalt and mineral aggregate, as is the case with typical composite systems. Adhesion work is the energy required per unit area to separate asphalt from aggregate, and it can be used to assess the adhesion between these materials.

The adhesion work ($W_{adhesion}$) has been determined by means of the interaction energy ($\Delta E_{asphalt\text{-}aggregate}$) defined in Eq. (4). The positive number for $W_{adhesion}$ means an attractive among the two parts, whereas the opposite result illustrates repulsion.

$$W_{adhesion} = \frac{-\Delta E_{asphalt-aggregate}}{A} = \frac{-(E_{total} - E_{asphalt} - E_{aggregate})}{A}$$

(4)

where $W_{adhesion}$ and $\Delta E_{asphalt\text{-}aggregate}$ are the adhesion work and interaction energy for asphalt and stone aggregates, correspondingly. $E_{asphalt}$ and $E_{aggregate}$ are the potential energy for the separated asphalt and aggregates. A is the Connolly molecular dimension on the mineral surface. $E_{total}$ is the overall potential energy for the interface system.

Fig 7 illustrates that the mineral type had a substantial impact on the effect of age on adhesive properties. As $SiO_2$ is electrically neutral, van der Waals interactions played an essential part in asphalt adhesion; by contrast, electrostatic interactions were minimal or absent [22]. Aging increased the distance between the asphalt and $SiO_2$ layers (Fig 6), reducing the van der Waals energy [19]. For the alkali mineral $CaCO_3$, electrostatic interactions were the main contributor to asphalt adhesion [22]. The increases in oxygen atoms and polarity during aging increased the atomic charges, contributing to significant electrostatic interactions with $CaCO_3$ [19]. In summary, the aggregate substance had an essential impact on the consequence of aging on asphalt-aggregate adhesion. In addition, $SiO_2$ exhibited inferior adhesion to asphalt relative to $CaCO_3$, demonstrating that the alkaline minerals adhered to the slightly acidic asphalt with greater intensity than the acidic minerals. This phenomenon aligned with earlier experimental findings [51] and MD simulation results [17].

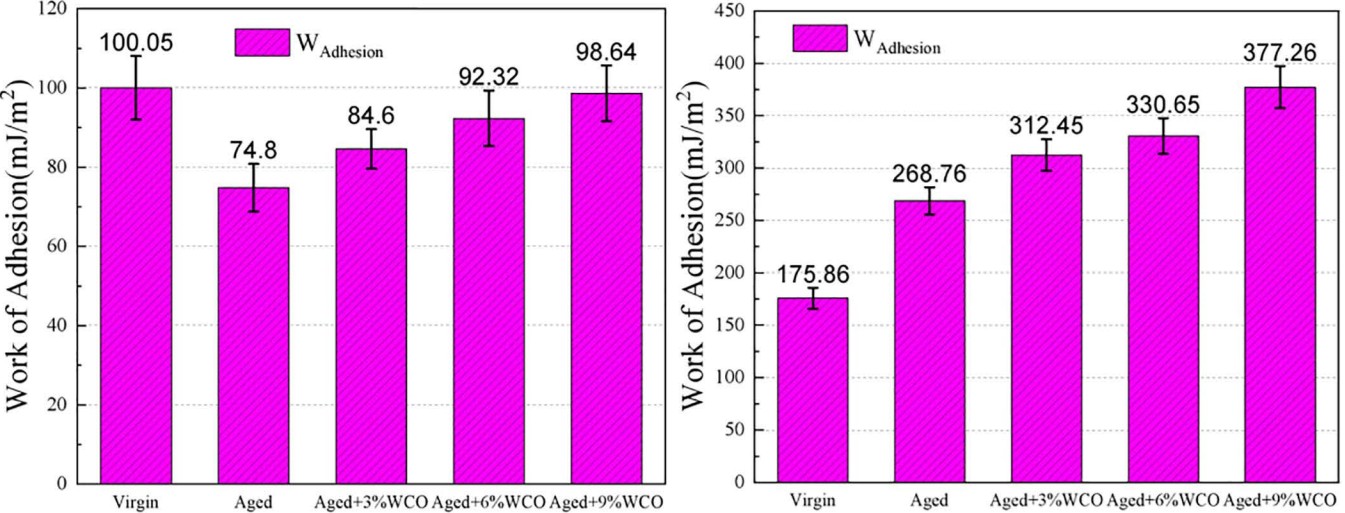

**Fig 7. Adhesion work between asphalt and aggregates under different asphalt conditions.**

The work of adhesion can be enhanced through the introduction of WCO, demonstrating that WCO strengthens the surface adherence of aged asphalt. The ability of WCO to breakdown the aggregate structure of aged asphalt, as well as the particular adhesion between WCO and aggregates, are the primary causes for improved adhesion [49].

**3.2.5. Influence of WCO on debonding work.** Moisture damage is a typical problem with asphalt pavements. It is caused by the invasion of dynamic water, which gradually degrades the adhesion across the asphalt-aggregate boundary and eventually leads to the peeling of the interface zone [50]. To gain greater knowledge of the implication of WCO on resistance to water at interfaces, the work of debonding proposed in prior literature was estimated [52,53]. According to Eq. (5), debonding work ($W_{debonding}$) was the quantity of energy necessary for water to separate from the asphalt mixture interface [31].

$$W_{debonding} = \frac{(\Delta E_{asphalt-water} + \Delta E_{aggregate-water} - \Delta E_{asphalt-aggregate})}{A}$$

(5)

where $W_{debonding}$ is the debonding work, $\Delta E_{asphalt-water}$ is the potential energy generated by asphalt-water coupling, $\Delta E_{aggregate-water}$ is the potential energy arising from the asphalt binder-water coupling, $\Delta E_{asphalt-aggregate}$ is the potential energy obtained from the asphalt mixture, $\Delta E_{aggregate-water}$ is the aggregate-water interaction energy, $\Delta E_{asphalt-aggregate}$ is the asphalt-aggregate interaction energy, and A is the interface contact area.

A negative $W_{debonding}$ indicated that energy can be released during deposition, signifying that the process operates naturally without using external energy. Thus, the resistance of the asphalt mixture to water damage diminished as the amount of released energy increased.

Fig 8 depicts the $W_{debonding}$ of various asphalt-aggregate models. All models had negative debonding work, demonstrating that the separation of the asphalt binders and aggregates by moisture occurred naturally without needing to absorb energy. Water would have conveniently permeated the asphalt-aggregate connection attributed to the hydrophobic characteristics to the asphalt and the hydrophilic features for the aggregates.

As the asphalt aging, the debonding work for both aggregates increased, demonstrating that the aged asphalt was more susceptible to water than the primative asphalt. Incorporating WCO minimizes the debonding work of aged asphalt

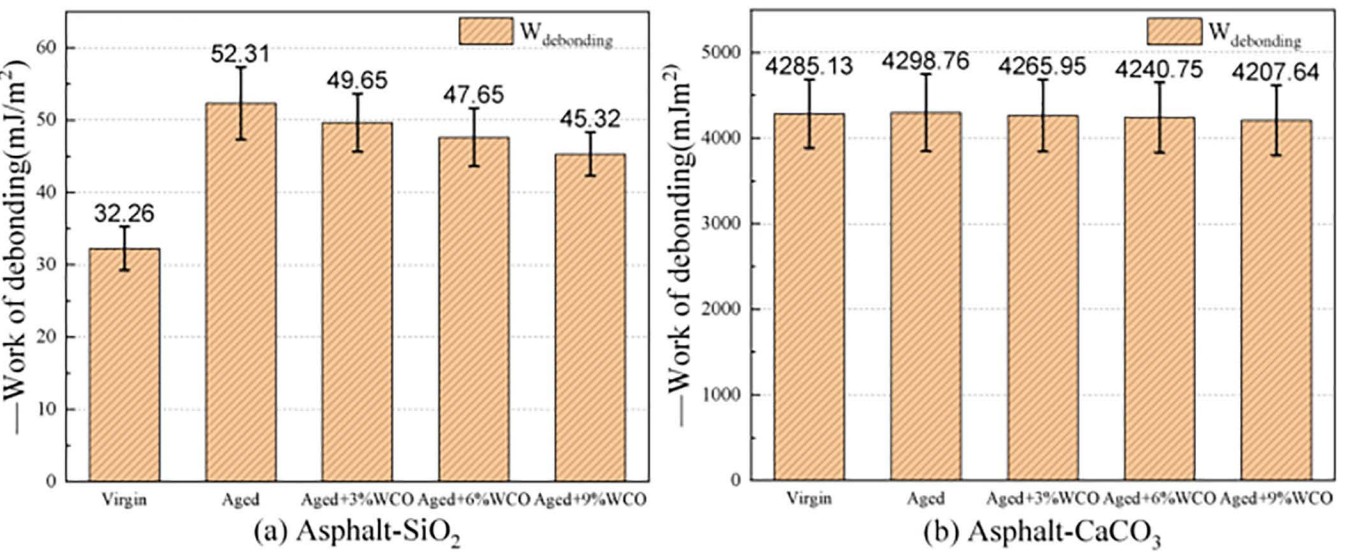

**Fig 8. Debonding work between asphalt and aggregates for various asphalt types.**

while strengthening water resistance. Furthermore, the debonding energy within the asphalt-water-$CaCO_3$ interface appears to be greater than the one as the asphalt-water-$SiO_2$ interface, demonstrating that water could more easily remove $CaCO_3$ from the asphalt mixture.

**3.2.6. Consequences of WCO upon moisture susceptibility.** The moisture sensitivity of the asphalt-aggregate interface might be determined by calculating the energy ratio (ER) of the adhesion and debonding work (Eq. (6)). An asphalt aggregate with a high ER value was deemed exceptionally resistant to water erosion.

$$ER = \left| W_{adhesion} / W_{debonding} \right|$$

(6)

Fig 9 depicts the ER results for several asphalt-aggregate interface configurations. The ER level at the asphalt-$SiO_2$ surface was found to be superior to that at the asphalt-$CaCO_3$ interfaces, indicating that the mixture of asphalt and $SiO_2$ possesses stronger water resistance than asphalt-$CaCO_3$. The water resistance of the asphalt-$SiO_2$ interactions deteriorates during aging process while the asphalt-$CaCO_3$ interface strengthens. The ER improvement of the WCO-rejuvenated asphalt demonstrated that WCO enhanced the water resistance of the aged asphalt to a certain extent. The computational results aligned with the test data, demonstrating that the WCO-rejuvenated asphalt had exceptional resistance to water damage [54].

## 3.3. Relationship between macro- and micro-evaluation results

The degree of adhesion for the asphalt-aggregate interaction might be determined through both pull-off strength test results (Section 3.1) and bonding work (Section 3.2.4). Their correlations were plotted in Fig 10.

Fig 10 indicates a considerable positive correlation between bond strength and interface adhesion work, confirming the coherence of macro and micro-evaluation methodologies for interface state.

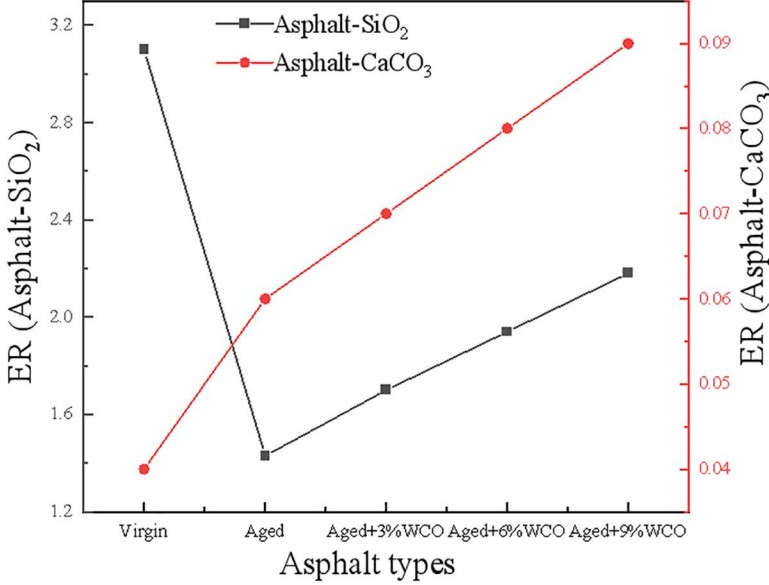

**Fig 9. Energy ratios for various asphalt–aggregate interfacial models.**

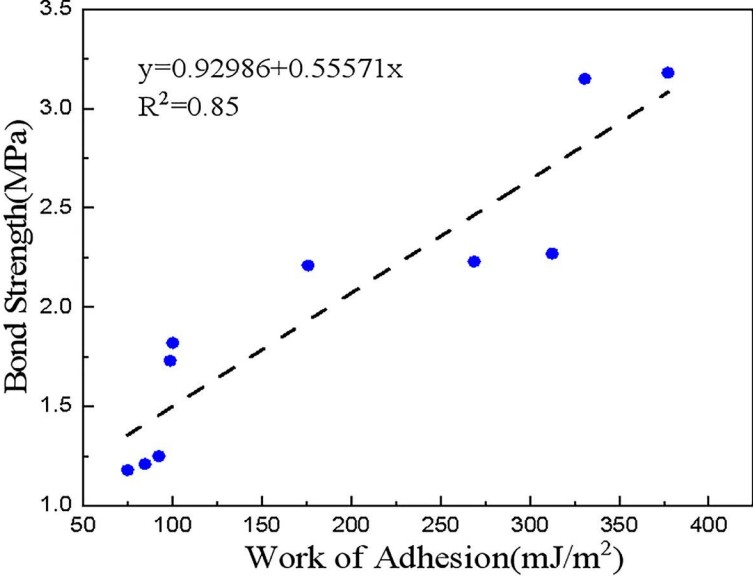

**Fig 10.  Correlation between bond strength and interfacial adhesion work.**

## 4. Conclusions

In this work, MD simulations and bond strength tests were combined to examine the adhesive performance of asphalt-aggregate interfaces rejuvenated using high-acid-value WCO at multiple scales. Furthermore, correlation analysis was performed between the simulated and experimental data. The findings are as follows:

(1)  The pull-off test results revealed that WCO may enhance the binding property of aged asphalt to a certain extent. It becomes approximately the level of virgin asphalt when the WCO adding amount was 9%.

(2)  The addition of WCO to the aged asphalt improved asphaltene self-aggregation and SARA component distribution, thus partially restoring the microscopic structure of the aged asphalt.

(3)  Mobility of SARA components in the aged asphalt binder could be enhanced by adding WCO. Asphalt components migrate more readily on the surface of $SiO_2$ than on the face of $CaCO_3$, and the powerful connection between asphalt and alkaline minerals hampers the movement of those molecules.

(4)  The type of aggregate exhibited an impact on the rejuvenating impacts of WCO on the aged asphalt-aggregate unity.

## Author contributions

**Conceptualization:** Zhiyu Wang, Kunjie Li, Xiaodong Huo.

**Data curation:** Zhiyu Wang, Qiang Pei, Xiaodong Huo, Yongwei Wang, Yanfang Li.

**Formal analysis:** Kunjie Li, Yanfang Li.

**Investigation:** Zhiyu Wang, Qiang Pei, Kunjie Li, Zhonghui Wang, Yongwei Wang, Shaoqi Kong.

**Methodology:** Zhiyu Wang, Qiang Pei, Zhonghui Wang, Yongwei Wang, Shaoqi Kong.

**Project administration:** Kunjie Li, Yongwei Wang.

**Software:** Shaoqi Kong.

**Supervision:** Zhonghui Wang, Xiaodong Huo.

**Validation:** Yanfang Li, Shaoqi Kong.

**Visualization:** Shaoqi Kong.

**Writing – original draft:** Zhiyu Wang, Qiang Pei, Kunjie Li, Zhonghui Wang, Xiaodong Huo, Yongwei Wang, Yanfang Li.

**Writing – review & editing:** Zhiyu Wang, Qiang Pei, Kunjie Li, Zhonghui Wang, Xiaodong Huo, Yongwei Wang, Yanfang Li.

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
