## [Decision Letter · Decision Letter 0]

31 Jul 2025

Dear Dr. wang,

Thank you for submitting your manuscript to PLOS ONE. After careful consideration, we feel that it has merit but does not fully meet PLOS ONE’s publication criteria as it currently stands. Therefore, we invite you to submit a revised version of the manuscript that addresses the points raised during the review process.

We look forward to receiving your revised manuscript.

Kind regards,

Jiaolong Ren

Academic Editor

PLOS ONE

Journal Requirements:

“The authors acknowledge support from the Luliang City High-Level Scientific and Technological Talent Introduction Key Research and Development Project (2022RC17), Luliang City high-tech field key research and development projects (2024GX06, 2024GX07), Shanxi Province Science Foundation for Youths Award (202303021222253), and the Innovative Development Plan Project of Shanxi Transportation Research Institute Group Co., Ltd. (22-JKCF-07). “

“The authors acknowledge support from the Luliang City High-Level Scientific and Technological Talent Introduction Key Research and Development Project (2022RC17), Luliang City high-tech field key research and development projects (2024GX06, 2024GX07), Shanxi Province Science Foundation for Youths Award (202303021222253), and the Innovative Development Plan Project of Shanxi Transportation Research Institute Group Co., Ltd. (22-JKCF-07). “

“The author(s) received no specific funding for this work”

Reviewers' comments:

Reviewer's Responses to Questions

**Comments to the Author**

1. Is the manuscript technically sound, and do the data support the conclusions?

Reviewer #1: Yes

Reviewer #2: Yes

2. Has the statistical analysis been performed appropriately and rigorously?

Reviewer #1: No

Reviewer #2: Yes

3. Have the authors made all data underlying the findings in their manuscript fully available?

Reviewer #1: Yes

Reviewer #2: Yes

4. Is the manuscript presented in an intelligible fashion and written in standard English?

Reviewer #1: Yes

Reviewer #2: Yes

Reviewer #1: The paper presents valuable findings, but language, figure quality, simulation clarity, and discussion depth must be improved before the manuscript is suitable for publication.

Lines 17–19: "Although the affects of aging on asphalt interface adhesive depend on mineral category..." → Replace “affects” with “effects”.

Lines 94–96: “...which is identical to the lightweight ingredients in virgin asphalt...” change the sentence for better understanding

Lines 156, 165, 428, 579: Multiple instances of awkward phrasing and grammar errors exist throughout the manuscript.

Lines 125–136: The literature gap is not very well defined, as the similar work is already done by e.g., Yan et al. 2022; Gao et al. 2019

Figures 3, 5, 7, 8, 10 (Lines ~345–600): The y-axis labels on several graphs (e.g., “Bond strength”, “Work of adhesion”) are too small or unclear. Also, error bars are missing, which are important for interpreting variability across replicates (especially in pull-off strength tests – Fig. 3). Revise figures for better visual clarity and inclusion of standard deviations.

Lines 315–320 and 607–609: The units for adhesion work and debonding (mJ/m²) should be consistently formatted. Also, in equations (1)–(6), symbols should be clearly defined immediately after the equation, not included in the paragraphs.

Equations (1)–(6), Lines ~370–590: The formatting of equations in text is inconsistent (e.g., spacing issues, improper symbols).

Lines 525–540 and 566–575: The points about WCO improving asphaltene dispersion and microstructure are repeated.

Lines 660–675: Conclusions are well-structured but could benefit from quantitative mention of key improvements (e.g., % increase in bond strength or adhesion energy after WCO addition).

Reviewer #2: The article is sound enough for publishing as it expose gap of knowledge and follows good methodology to attain the objectives.

Comments:

1. The introduction must be edited to focus on the problem statement, objectives and research gap. Avoid the general and repeated speech.

2. Some figures are presented with low quality missing axis. Double check and fix.

3. The discussions must be deepened. The results must be justified, verified, and validated.

4. The English must be edited.

**Do you want your identity to be public for this peer review?** For information about this choice, including consent withdrawal, please see our Privacy Policy

Reviewer #1: No

Reviewer #2: **Yes: ** Prof. Dr. Ahmed Mancy Mosa

---

## [Author Response · Author response to Decision Letter 1]

25 Aug 2025

Journal Requirements:

Thank you for the valuable suggestions provided by the editor. Based on the editor's advice, the article has been revised accordingly.

Thank you for the valuable suggestions provided by the editor. Based on the editor's advice, the article has been revised accordingly.

Thank you for the valuable suggestions provided by the editor. Based on the editor's advice, the article has been revised accordingly.

“The authors acknowledge support from the Luliang City High-Level Scientific and Technological Talent Introduction Key Research and Development Project (2022RC17), Luliang City high-tech field key research and development projects (2024GX06, 2024GX07), Shanxi Province Science Foundation for Youths Award (202303021222253), and the Innovative Development Plan Project of Shanxi Transportation Research Institute Group Co., Ltd. (22-JKCF-07). “

Thank you for the valuable suggestions provided by the editor. Based on the editor's advice, the article has been revised accordingly.

“The authors acknowledge support from the Luliang City High-Level Scientific and Technological Talent Introduction Key Research and Development Project (2022RC17), Luliang City high-tech field key research and development projects (2024GX06, 2024GX07), Shanxi Province Science Foundation for Youths Award (202303021222253), and the Innovative Development Plan Project of Shanxi Transportation Research Institute Group Co., Ltd. (22-JKCF-07). “

“The author(s) received no specific funding for this work”

Thank you for the valuable suggestions provided by the editor. Based on the editor's advice, the article has been revised accordingly.

Reviewers' comments:

Reviewer's Responses to Questions

Comments to the Author

1. Is the manuscript technically sound, and do the data support the conclusions?

Reviewer #1: Yes

Reviewer #2: Yes

2. Has the statistical analysis been performed appropriately and rigorously?

Reviewer #1: No

Reviewer #2: Yes

3. Have the authors made all data underlying the findings in their manuscript fully available?

Reviewer #1: Yes

Reviewer #2: Yes

4. Is the manuscript presented in an intelligible fashion and written in standard English?

Reviewer #1: Yes

Reviewer #2: Yes

5. Review Comments to the Author

Reviewer #1: The paper presents valuable findings, but language, figure quality, simulation clarity, and discussion depth must be improved before the manuscript is suitable for publication.

Lines 17–19: "Although the affects of aging on asphalt interface adhesive depend on mineral category..." → Replace “affects” with “effects”.

Thank you to the reviewers for the valuable suggestions. According to the reviewer's opinion, it has been corrected in lines 17 to 19 of the paper.

Lines 94–96: “...which is identical to the lightweight ingredients in virgin asphalt...” change the sentence for better understanding

Thank you to the reviewers for the valuable suggestions. According to the reviewer's opinion, it has been corrected in lines 94 to 96 of the paper.

Lines 156, 165, 428, 579: Multiple instances of awkward phrasing and grammar errors exist throughout the manuscript.

Thank you to the reviewers for the valuable suggestions. According to the reviewer's opinion, it has been corrected in the paper.

Lines 125–136: The literature gap is not very well defined, as the similar work is already done by e.g., Yan et al. 2022; Gao et al. 2019

Thank you to the reviewers for the valuable suggestions. According to the reviewer's opinion, it has been summarize again in the paper.

Figures 3, 5, 7, 8, 10 (Lines ~345–600): The y-axis labels on several graphs (e.g., “Bond strength”, “Work of adhesion”) are too small or unclear. Also, error bars are missing, which are important for interpreting variability across replicates (especially in pull-off strength tests – Fig. 3). Revise figures for better visual clarity and inclusion of standard deviations.

Thank you to the reviewers for the valuable suggestions. According to the reviewer's opinion, figures 3, 5, 7, 8, 10 has been revised in the paper.

Lines 315–320 and 607–609: The units for adhesion work and debonding (mJ/m²) should be consistently formatted. Also, in equations (1)–(6), symbols should be clearly defined immediately after the equation, not included in the paragraphs.

Equations (1)–(6), Lines ~370–590: The formatting of equations in text is inconsistent (e.g., spacing issues, improper symbols).

Thank you to the reviewers for the valuable suggestions. According to the reviewer's opinion, figures 8, 10 and equations (1)–(6) has been revised in the paper.

Lines 525–540 and 566–575: The points about WCO improving asphaltene dispersion and microstructure are repeated.

Thank you very much for the valuable comments provided by the reviewers. Based on their suggestions, we have made corresponding revisions in the paper.d

Lines 660–675: Conclusions are well-structured but could benefit from quantitative mention of key improvements (e.g., % increase in bond strength or adhesion energy after WCO addition).

Thank you to the reviewers for the valuable suggestions. According to the reviewer's opinion, relevant conclusions has been revised in the paper.

Reviewer #2: The article is sound enough for publishing as it expose gap of knowledge and follows good methodology to attain the objectives.

Comments:

1. The introduction must be edited to focus on the problem statement, objectives and research gap. Avoid the general and repeated speech.

Thank you to the reviewers for the valuable suggestions. According to the reviewer's opinion, it has been corrected in the paper.

2. Some figures are pr esented with low quality missing axis. Double check and fix.

Thank you to the reviewers for the valuable suggestions. According to the reviewer's opinion, it has been corrected in the paper.

3. The discussions must be deepened. The results must be justified, verified, and validated.

Thank you to the reviewers for the valuable suggestions. According to the reviewer's opinion, it has been corrected in the paper.

4. The English must be edited.

Thank you to the reviewers for the valuable suggestions. According to the reviewer's opinion, it has been corrected in the paper.

---

## [Decision Letter · Decision Letter 1]

4 Sep 2025

Comprehensive analysis of effect of high-acid-value waste cooking oil on adhesion between aged asphalt and aggregates

PONE-D-25-36592R1

Dear Dr. Wang,

We’re pleased to inform you that your manuscript has been judged scientifically suitable for publication and will be formally accepted for publication once it meets all outstanding technical requirements.

Kind regards,

Jiaolong Ren

Academic Editor

PLOS ONE

Additional Editor Comments (optional):

Reviewer #1:

Reviewer #2:

Reviewers' comments:

Reviewer's Responses to Questions

**Comments to the Author**

Reviewer #1: All comments have been addressed

Reviewer #2: All comments have been addressed

2. Is the manuscript technically sound, and do the data support the conclusions?

Reviewer #1: Yes

Reviewer #2: Yes

3. Has the statistical analysis been performed appropriately and rigorously?

Reviewer #1: Yes

Reviewer #2: Yes

4. Have the authors made all data underlying the findings in their manuscript fully available?

Reviewer #1: Yes

Reviewer #2: Yes

5. Is the manuscript presented in an intelligible fashion and written in standard English?

Reviewer #1: Yes

Reviewer #2: Yes

Reviewer #1: (No Response)

Reviewer #2: The authors considered all comments and made the suitable corrections. Good luck. ……………………………………………………

**Do you want your identity to be public for this peer review?** For information about this choice, including consent withdrawal, please see our Privacy Policy

Reviewer #1: No

Reviewer #2: **Yes: ** Prof.Dr. Ahmed Mancy Mosa

---

## [Editor Report · Acceptance letter]

PONE-D-25-36592R1

PLOS ONE

Dear Dr. wang,

I'm pleased to inform you that your manuscript has been deemed suitable for publication in PLOS ONE. Congratulations! Your manuscript is now being handed over to our production team.

Kind regards,

on behalf of

Dr. Jiaolong Ren

Academic Editor

PLOS ONE